# Evidence and Metabolic Implications for a New Non-Canonical Role of Cu-Zn Superoxide Dismutase

**DOI:** 10.3390/ijms24043230

**Published:** 2023-02-06

**Authors:** Ziqiao Sun, Xin-Gen Lei

**Affiliations:** Department of Animal Science, Cornell University, Ithaca, NY 14853, USA

**Keywords:** copper/zinc, SOD1, YWHAZ (14-3-3 ζ), YWHAE (14-3-3 ε), protein–protein interaction

## Abstract

Copper–zinc superoxide dismutase 1 (SOD1) has long been recognized as a major redox enzyme in scavenging superoxide radicals. However, there is little information on its non-canonical role and metabolic implications. Using a protein complementation assay (PCA) and pull-down assay, we revealed novel protein–protein interactions (PPIs) between SOD1 and tyrosine 3-monooxygenase/tryptophan 5-monooxygenase activation protein zeta (YWHAZ) or epsilon (YWHAE) in this research. Through site-directed mutagenesis of SOD1, we studied the binding conditions of the two PPIs. Forming the SOD1 and YWHAE or YWHAZ protein complex enhanced enzyme activity of purified SOD1 in vitro by 40% (*p* < 0.05) and protein stability of over-expressed intracellular YWHAE (18%, *p* < 0.01) and YWHAZ (14%, *p* < 0.05). Functionally, these PPIs were associated with lipolysis, cell growth, and cell survival in HEK293T or HepG2 cells. In conclusion, our findings reveal two new PPIs between SOD1 and YWHAE or YWHAZ and their structural dependences, responses to redox status, mutual impacts on the enzyme function and protein degradation, and metabolic implications. Overall, our finding revealed a new unorthodox role of SOD1 and will provide novel perspectives and insights for diagnosing and treating diseases related to the protein.

## 1. Introduction

Copper–zinc superoxide dismutase 1 (SOD1 or Cu-Zn-SOD) is present mainly in the cytoplasm, with a small portion in the nucleus, of nearly all eukaryotes [1]. As a crucial component of cellular defense against oxidative stress, SOD1 breaks down O_2_^•−^ through a redox cycling of the copper ion in the Cu-Zn active site to dismutate O_2_^•−^ to H_2_O_2_ and O_2_ [2]. In addition to this enzymatic role in free radical scavenging, SOD1 has recently been unraveled with unconventional roles. Lu et al. discovered SOD1 as an RNA binding protein or regulator of RNA stability, in which mutant SOD1 (G93A) specifically altered the stability of ribonucleoprotein complex associated with 3′UTR mRNA of vascular endothelial growth factor but not wild-type SOD1 [3,4]. Tsang’s group revealed another novel, unorthodox role of SOD1 in nuclear translocation. In the nucleus, phosphorylated SOD1 bound genomic DNA promoter and mediated the expression of genes in oxidative resistance and DNA repair, which was regulated by the ATM/Mec1 oxidative sensor in response to increased H_2_O_2_ [5]. Liu’s group recently discovered that SOD1 could bind double-stranded DNA (dsDNA) by small-angle X-ray scattering, and that the SOD1-dsDNA complex responded to different oxidants by changing its shape, providing a new potential mechanism for SOD1 to protect cells from oxidative stress [6]. SOD1 could also function as a protein interactor with copper chaperone for SOD1 (CCS) to regulate itself maturation, dimerization, and activity [7,8]. Loss of SOD1 protein not only led to increased reactive oxygen species (ROS) levels and oxidative damage [9,10,11,12] but also induced resistance to acetaminophen toxicity, defects in femoral mechanical performance, injury of islets beta-cell function, and exacerbated lipogenesis with elevated non-esterified fatty acids (NEFA), total cholesterol (TC), and triglycerides (TG) [13,14,15,16,17]. Because some of these phenotypes were not readily explained by the SOD1 enzyme activity loss *per se*, there might be unrevealed, unorthodox roles of SOD1 involved in these abnormalities.

Tyrosine 3-monooxygenase/tryptophan 5-monooxygenase activation proteins zeta (YWHAZ) and epsilon (YWHAE) belong to the 14-3-3 protein family that has other five isoforms (YWHAB, YWHAG, YWHAH, YWHAQ, and YWHAS). These proteins are ubiquitously produced in human tissues, with the highest abundance in the brain [18]. They are widely regarded as molecular adapters modulating over 200 diverse signaling proteins, especially in brain neuronal development and neurological disorders [19,20]. The multifunctions of 14-3-3 family proteins make them hard to study, due to a likely complicated network complex instead of a simple role in a single pathway. YWHAZ protein is a crucial regulator of cell growth and apoptosis pathways [21,22] and is closely related to several types of cancer [23,24]. Additionally, knockout of YWHAZ (14-3-3ζ) led to conspicuously lean in mouse pups and diminished visceral adipose accumulation, while overexpression of YWHAZ induced obesity-like phenotypes [25]. Loss of YWHAZ also increased fasting insulin levels with unaltered beta-cell glucose sensitivity [26]. However, specific functions of YWHAE (14-3-3ε) in metabolism remain unclear, although it was implicated with a latent role in insulin signaling as it interacted with insulin-like growth factor I receptor and insulin receptor substrate I [27].

Putative protein–protein interactions (PPIs) between SOD1 and YWHAE or YWHAZ, along with other two-family members (YWHAQ and YWHAG), were proposed based on a pool of protein pairs identified by affinity purification mass spectrometry (AP/MS) [28,29]. However, the AP/MS analysis results were confounded with a high background of non-specific protein binders [30]. To overcome this constraint, we applied the specific protein complementation assay (PCA) followed by GST pull-down assay to provide the first direct evidence for these PPIs. Moreover, we prepared several SOD1 single-site mutants (A4V, H46R, G85R, and G93A) associated with amyotrophic lateral sclerosis (ALS) disease [31,32,33,34] and another mutation D124N in the SOD1 zinc active site [35] to compare disruptions of the PPIs between SOD1 and YWHAE or YWHAZ at different levels of remaining SOD1 enzymatic activities. Subsequently, we overexpressed and purified the three proteins and determined the binding conditions and effects of their PPIs on stability of the three proteins and the activity of SOD1. Thereafter, we explored if these PPIs modulated lipid metabolism, cell growth, and cell survival of HEK293T cells. Altogether, we discovered two new PPIs between SOD1 and YWHAZ or YWHAE and revealed a novel, unorthodox role of SOD1 beyond its well-known redox function.

## 2. Results

### 2.1. Evidence for Protein–Protein Interactions between SOD1 and YWHAE or YWHAZ

We performed PCA to determine the postulated PPIs between human SOD1 with YWHAE or YWHAZ in HEK293T cells. The principle of PCA (Figure 1A) is that bait and prey proteins are fused to two complementary fragments (F1 and F2) of a yellow fluorescent protein (YFP) [36]. If the bait and prey proteins interact, the two fragments will covalently link to generate a yellow fluorescent signal. We prepared the plasmid constructs in four orientations to measure each protein–protein pair as F1 or F2 tagged in the N (F1N or F2N) or C (F1C or F2C) terminal of the two proteins (Figure 1B). The fluorescence signal intensity was normalized to the cell group transfecting the vector control in the same orientation. The YWHAZ and YWHAE had pronounced fluorescence signals compared with a negative control protein ATPG (fold change close to 1 and almost no fluorescence signal under the microscope) that was not supposed to interact with SOD1 from our preliminary screening (Figure 1C). Relatively, the strongest interaction of SOD1 with YWHAE occurred in the N-N orientation (17.4-fold, *p* < 0.01) and with YWHAZ in the C-N orientation (3.72-fold, *p* < 0.001) compared with the same orientation of ATPG negative control. Subcellular location and relative intensity of the fluorescent cell imaging suggested cytosol as the primary site for these PPIs, along with moderate occurrence in the nucleus.

We expressed mouse proteins in human HEK293T for PCA to test whether the PPIs were independent of species. Whereas YWHAE and YWHAZ sequences are highly homologous between humans and mice, their SOD1 sequences have only 84% identity (Appendix A). The PCA results indicated that the PPIs were shown in the C-N orientation of mSOD1-mYWHAE (2.06-fold, *p* < 0.01) and in the N-C orientation of mSOD1-mYWHAZ (39.1-fold, *p* < 0.001) compared with the same orientation of ATPG negative control (Appendix A). These results proved the PPIs existed between SOD1 and YWHAE or YWHAZ and indicated that the PPIs were relatively independent and could be cross-species.

### 2.2. Protein Complex between the Purified SOD1 and YWHAE or YWHAZ

To determine if SOD1-YWHAZ and SOD1-YWHAE could form protein complexes, we used *Pichia Pastrois X33* to express the His-tag conjugated SOD1-His and YWHAZ-His and *E. coli BL21 (DE3)* to express the GST-conjugated GST-YWHAE and GST-YWHAZ. As shown in Figure 2A, GST-YWHAE and GST-YWHAZ could bind and pulled-down SOD1-His, despite a relatively low amount of the complex. Because GST-YWHAZ and GST-YWHAE proteins were expressed intracellularly, certain non-specific proteins remained after the purification might interfere with the interaction in the molecular weight region of SOD1. To confirm the target protein bands, we performed immunoblotting against the SOD1 and YWHAZ antibodies (Figure 2B). While setting the SOD1 protein as constant, different ratios of input proteins affected the complex formation, where increasing YWHAZ or YWHAE protein facilitated the complex formation (Figure 2C). As the large molecular size of the GST tag might impede the PPI, we tested YWHAZ tagged to only six histidines (YWHAZ-His, expressed and purified from *Pichia Pastrois X33*). Due to both proteins having his tag, the purified SOD1 was conjugated with a SOD1 antibody and immobilized on agarose beads G for pull down assay. The YWHAZ-His protein was detected from the immunoprecipitation eluates on lane 3 (Figure 2D), indicating the PPI between YWHAZ and SOD1 and in vitro protein complex formation. Summarizing all evidence, we could confirm the formation of protein complexes between SOD1 and YWHAE or YWHAZ.

### 2.3. Impacts of SOD1 Mutations and Oxidative Stress on Protein–Protein Interactions between SOD1 and YWHAE or YWHAZ

To determine impacts of SOD1 mutations on the focused PPIs, we performed PCA using two-orientated (F1N-F2N, F1C-F2N) constructs of YWHAE or YWHAZ with the wild-type (WT) or mutants of human SOD1. The upper part of Table 1 shows the absolute fold changes in fluorescence intensity for the interactions of SOD1 mutants with YWHAE or YWHAZ. The lower sectiont of Table 1 converts the fluorescent intensity fold changes to the percentage of remaining PPI relative to that of the WT-SOD1 by method Formula (1). The transfection efficiency of various plasmids was assumed at similar levels (Appendix A). Compared with the WT, four SOD1 mutants (H46R, G85R, G93A, D124N) had decreased affinity to interact with YWHAE and YWHAZ proteins by over 50% in either orientation. However, overexpressing most of these mutants of SOD1 did not increase total intracellular SOD1 activity, but some mutants resulted in a moderate decrease (up to 30%, *p* < 0.05) in the activity (Appendix A). Other studies showed similar decreases in total SOD1 activities (0–45% to wild-type SOD1) among these purified SOD1 mutant proteins [31,32,37,38]. However, the impaired or disrupted PPIs were largely independent of total intracellular SOD1 activity. Notably, the strongest breakdown in the PPIs mediated by the mutated SOD1 was different between the interactions with YWHAE and YWHAZ protein. The SOD1-D124N mutant decreased the interaction with YWHAE by 96.0% in F1C-F2N orientation (*p* < 0.01), whereas the SOD1-G85R mutant reduced the interaction with YWHAZ by 99.7% in F1N-F2N orientation (*p* < 0.05). Plausibly, there were different binding conditions of SOD1 with YWHAE and YWHAZ.

To determine impacts of redox status on the PPI of SOD1 with YWHAE (as a representative), we treated the transfected cells with various oxidants and antioxidants. The treatments of diquat (a superoxide generator, 50 μM) and TBHP (a hydroperoxide generator, 50 μM) attenuated the interaction at 1 h and 2 h (1h: 16% and 28% and 2h: 11% and 20%, respectively, *p* < 0.05, Appendix A). However, those effects became insignificant after a longer incubation, which made us speculate that the effects of diquat and TBHP could be just acute. Among three ROS inhibitors (50 μM), NAC (a GSH precursor in glutathione elevation biosynthesis) slightly rescued the SOD1-H46R (as a representative) and YWHAE interaction (Appendix A), while CuDIPs (a SOD mimic) almost blocked the interaction formation, and ebselen (a GPX mimic) made no significant effect. The same effect trend was also observed in the SOD1-D124N and YWHAE interaction (Appendix A). In the same way that oxidants impaired PPIs, the disrupted PPIs between SOD1 mutants (H46R and D124N) and YWHAE could be restored by antioxidant compensation.

### 2.4. Impacts of Protein–Protein Interactions between SOD1 and YWHAE or YWHAZ on SOD1 Activity

Incubating purified SOD1 protein with purified YWHAE protein or YWHAZ protein in the SOD1 activity assay buffer for 4 h, at 4 °C, increased the SOD1 activity by 40% compared with SOD1 plus buffer group (*p* < 0.05, Figure 3A). The SOD1 activity kept increasing over the incubation time. A 15 h incubation with the YWHAZ protein at a ratio of 1:3 (1 SOD1 to 3 YWHAZ protein amount) caused the highest elevation of SOD1 activity (~2-fold increase, *p* < 0.01) (Figure 3B). To verify if such enhancement could be replicated in cells, we transfected the YWHAZ vector into HEK293T and found a 20% increase (*p* < 0.05) in SOD1 activity (Figure 3C). We also generated YWHAZ-knockout (YWHAZ-KO) and SOD1-knockdown-YWHAZ-knockout (SOD1-KD&YWHAZ-KO) cells by CRISPR-Cas9 genome-editing technology (Appendix A). The YWHAZ-KO cells had a 15.2% decrease (*p* < 0.05) in SOD1 activity (Figure 3D) but unchanged SOD2 activity or SOD1 protein level (Appendix A). Inactivating the *YWHAZ* gene on top of SOD1 knockdown (SOD1-KD&YWHAZ-KO cells) showed an additional decrease in SOD1 activity over the SOD1-KD cells without affecting SOD2 activity (Figure 3E). Both in vitro and in vivo experiments illustrated that PPIs between SOD1 and YWHAE or YWHAZ could specifically promote the activity of SOD1.

### 2.5. Impacts of Protein–Protein Interactions between SOD1 and YWHAE or YWHAZ on Their Protein Stability

To determine effects of PPIs between SOD1 and YWHAE or YWHAZ on their relative turnover or resistance to protein degradation, we performed a cycloheximide-mediated protein degradation assay and a series of immunoblots in HEK293T cells transfected with plasmids expressing SOD1, YWHAE, and YWHAZ codon sequence. The SOD1 stability in HEK293T cells was increased by co-expressing the WT-SOD1 with YWHAE (17.7%, *p* < 0.01) or YWHAZ (13.5%, *p* < 0.05) (Figure 4A,B). Meanwhile, the protein stability of YWHAE (Figure 4A,C) and YWHAZ (Figure 4A,D) was increased by 40.4% (*p* < 0.01) and 14% (*p* < 0.01), respectively, via co-expressing WT-SOD1 with respective gene. The SOD1 mutants (G85R and D124N) had lower stability than WT-SOD1 (Figure 4F). However, the protein stability of SOD1 mutants (G85R and D124N) was unchanged by the co-expression with YWHAE (Figure 4E,F). Likewise, co-expressing SOD1 mutants (G85R and D124N) with YWHAE, which showed disrupted PPIs, did not improve their protein stability (Figure 4E,G). Strikingly, overproducing all three proteins (SOD1, YWHAE, and YWHAZ) simultaneously in the cells removed the protein stability benefits resulted from the co-expression of SOD1 with YWHAE or YWHAZ (Figure 4B–D). Overall, we identified that both proteins increased stability or were more resistant to their intracellular protein degradation when having a PPI with each other.

### 2.6. Impacts of Protein–Protein Interactions between SOD1 and YWHAE or YWHAZ on Lipid Metabolism and Relative Gene Expression

Both SOD1 and YWHAZ have shown important roles in lipid metabolism [16,17,25]. To investigate impacts of the focused PPIs on lipid metabolism, we used genome editing to generate SOD1-KD (SOD1 knockdown) and SOD1-KD&YWHAZ-KO HEK293T cells (SOD1 knockdown and YWHAZ knockout) (Appendix A). Knocking down SOD1 increased cellular lipid droplet accumulation, while knockout of YWHAZ protein restored the status (Figure 5A,B). Treating the SOD1KO cells with the SOD1 mimic (CuDIPs) did not prevent the abnormal lipid accumulation, but increased (by 1μM CuDIPs) the lipid accumulation (30%) when additional knocked out *YWHAZ* gene in SOD1-KD cells (SOD1-KD&YWHAZ-KO, Figure 5C). In searching for a plausible mechanism, we found that *SREBP1* (sterol regulatory element-binding protein 1) and *ACACA* (acetyl-CoA carboxylase alpha) were upregulated, while *LPL* (lipoprotein lipase), *FASN* (fatty acid synthase), and *HMGCR* (HMG-CoA reductase) were downregulated in SOD1-KD cells. These changes could enhance lipogenesis and attenuate lipolysis (Figure 5D). Other related genes (*HMGCS1, HMGCS2, LIPE,* and *PPARG*) showed similar downregulations, but the changes were not statistically significant (Appendix A). Interestingly, expression of these genes was restored to the control levels (WT) by knocking out YWHAZ in SOD1KO cells (Figure 5D). The responses of *HMGCS1, PNPLA2, ACACA, FASN, LPL,* and *LIPE* mRNA levels to CuDIPs varied with different genotypes of cells (Appendix A).

We also measured lipid and fatty acids profiles in HepG2 cells overexpressing WT-SOD1 or SOD1 mutants (H46R and G93A as representatives). We first demonstrated that overexpressing WT-SOD1, SOD1-H46R, or SOD1-G93A could dominate the PPIs between endogenous YWHAE or YWHAZ with WT-SOD1 since the co-immunoprecipitation bands under overexpressing (Lanes 6–8, Figure 5E) were stronger in intensity than vector control F1N (Lane 5, Figure 5E). Compared with the vehicle control, transfecting SOD1 mutants (H46R and G93A) showed no effect on TC content in HepG2 cells. The SOD1-G93R significantly elevated the TG concentration, while the WT-SOD1 and the SOD1-H46R transfection moderately decreased TG in HepG2 cells. The SOD1-H46R and SOD1-G93A transfection in HepG2 cells decreased palmitic acid (C16:0), stearic acid (C18:0), and oleic acid (C18:1n-9) levels. In particular, the C18:1n-9 was eliminated by the SOD1-H46R transfection (Figure 5G). HepG2 cells tended to accumulate lipid droplets after transfecting mutant SOD1-H46R and SOD1-G93A (Figure 5F). The mRNA levels of *SREBP1, SREBP2, HMGCS2* (3-hydroxy-3-methylglutaryl-CoA synthase 2)*,* and *LPL* were decreased by transfecting SOD1-G93A, whereas *ACACA* was upregulated by overexpressing SOD1-G93A (Figure 5H). Changes in mRNA levels of *HMRCS1, HMRCR,* and *FASN* were not statistically significant (Appendix A) by overexpressing WT-SOD1 or mutant SOD1. Interestingly, transfecting YWHAE increased lipid droplets accumulation, and the increase was blocked by cotransfecting SOD1 and YWHAE (Figure 5I and Appendix A).

In summary, knocking down SOD1 in HEK293T increased lipid accumulation, along with upregulating several lipogenesis-related genes, which is similar to hepatic-steatosis-like phenotypes we found in Sod1^−/−^ mice [16]. When overexpressing the SOD1-G93A mutant, it turned to bind more YWHAE or YWHAZ, and the PPIs altered the fatty acid profiles and affected the gene expression to elevate lipolysis. We also discovered the potentially correlated roles between SOD1 and YWHAZ or YWHAE in mediating lipid metabolism since the increasing trend was rescued when additional knockout YWHAZ or co-overexpressing YWHAE.

### 2.7. Impacts of Protein–Protein Interactions between SOD1 and YWHAE or YWHAZ on Cell Growth and Survival

Because the 14-3-3 protein family is well known for affecting cell growth and death, including YWHAE and YWHAZ [21,22,39], potential roles of PPIs between SOD1 and YWHAE or YWHAZ could play in these pathways. We used CRISPR-editing to alter expressions of endogenous SOD1 and YWHAZ in HEK293T cells to investigate impacts of their PPIs on cell growth (proliferation) and survival. Knockdown of SOD1 (SOD1-KD) or producing the SOD1 mutant (SOD1-A4V, alanine to valine at the fourth position) decreased cell growth (Figure 6A), and cell survival was also reduced after removal of FBS from the culture media at indicated time points (Figure 6B). Overexpressing YWHAE and SOD1 alone or together in the WT or SOD1-KD cells rescued cell growth (Figure 6C), but not cell survival (Figure 6D). Overexpressing YWHAZ and SOD1 also gave a moderate restoration of cell growth (Figure 6E) in both WT and SOD1-KD cells, and only overexpressing YWHAZ improved cell survival in SOD1-KD cells (Figure 6F). Cell growth was enhanced by overexpressing SOD1 in the SOD1-KD&YWHAZ-KO cells (Figure 6G), but cell survival was decreased (Figure 6H). Similar to SOD1-KD cells, SOD1-A4V cells also had declined SOD1 activity and protein levels (Appendix A). Overexpressing YWHAE or YWHAZ and SOD1 alone or together in SOD1-A4V cells rescued cell growth (Appendix A, C) but not cell survival (Appendix A). Compared with WT controls, mRNA levels of *CDKN1B* (cyclin-dependent kinase inhibitor 1B) and *CCND1* (cyclin D1) were downregulated by knockdown of SOD1 (SOD1-KD vs. WT, Figure 6I) but rescued by an additional knockout of YWHAZ (SOD1-KD vs. SOD1-KD&YWHAZ-KO, Figure 6I). Overexpressing WT-SOD1 instead of SOD1 mutants enhanced *CDKN1B* and *CCND1* mRNA levels (Appendix A) that were also increased by the graded CuDIPs treatment (Appendix A). However, the latter effect disappeared by knocking down SOD1 and knocking out YWHAZ. This evidence suggests that the impaired SOD1 (KD or A4V mutant) slackened cell proliferation and accelerated cell death. Moreover, YWHAE and YWHAZ might interact with SOD1 to affect cell growth and survival, and the relative ratio of protein amounts of YWHAE and YWHAZ to SOD1 could be a critical determinant.

### 2.8. Expression Correlation between SOD1 and YWHAE or YWHAZ Protein

If two proteins interact and have a co-function or regulatory relationship, they may be correlated in protein expression profiles [40]. To find co-function or regulatory relationships for the putative PPIs between SOD1 and YWHAE or YWHAZ, we used a breast cancer quantitative proteome database [41] to analyze latent correlations of their protein expression profiles [41]. There was a positive correlation between SOD1 and YWHAE (r = 0.71, *p* < 0.01, Figure 7A), but a negative correlation between SOD1 and YWHAZ (r = −0.40, *p* < 0.02, Figure 7B). Consistent with the negative correlation between SOD1 and YWHAZ, knockout of *Sod1* in mice elevated (*p* < 0.05) YWHAZ protein levels in the liver, brown adipose tissue (BAT) and muscle (Appendix A), but did not significantly decrease (positive correlation) in the protein expression level of YWHAE (Appendix A). The enhanced protein levels of both YWHAE and YWHAZ in the brown adipose tissue, muscle, and/or liver, but not other tissues, of *Sod1*^−/−^ mice reinforce that the opposite correlations between SOD1 and YWHAE (+) vs. YWHAZ (−) in the progression of the human breast cancer represented a specific interdependence or interaction of these proteins.

## 3. Discussion

In this study, we have provided direct evidence for the PPIs of SOD1 with YWHAE and YWHAZ under in vivo and in vitro conditions. Although these PPIs were proposed based on prior analysis of protein pairs by affinity purification mass spectrometry (AP/MS) [26,27], our research represents a significant progress. Comparatively, our methods of PCA were more specific and reliable than the AP/MS analysis that was confounded with a high background of non-specific protein binders [28]. Convincingly, we demonstrated the structural and redox environment dependences of the PPIs. Disruptions of the PPIs by mutagenesis of the binding SOD1 (H46R, G85R, G93A, D124N) imply the importance of an intact SOD1 structure or involvements of these mutated sites in the PPIs. Additionally, the PPIs were readily impaired by external oxidants (diquat and TBHP). However, three antioxidants (NAC, CuDIPs, and ebselen) exhibited different potencies in rescuing disrupted PPIs between SOD1-H46R or SOD1-D124N with YWHAE, despite their common roles in suppressing oxidative stress [42,43]. Apparently, the PPIs between SOD1 and YWHAE or YWHAZ could be modulated by the intrinsic activity of SOD1 itself and exogenous antioxidants and oxidants in their own specific ways.

It is fascinating for us to unearth an enhanced SOD1 activity by the formation of protein complexes in the PPIs. The enhancement was proportional to the incubation time and YWHAE or YWHAZ protein inputs. Overexpressing or knocking out YWHAZ in HEK293T cells led to increased or decreased intracellular activity of SOD1, replicating the same in vitro impact of the PPIs on the SOD1 enzymatic function in vivo (cell). Based on 3D structure analysis of the PPI by AlphaFold2 [44], we have found that the constitutions of SOD1-YWHAE and SOD1-YWHAZ dimer complexes are highly similar (Appendix A). The high accuracy and confidence of this new advanced computational tool have been tested for predicting protein–protein interactions [45]. In dimer complexes, the YWHAE or YWHAZ dimer holds the SOD1 dimer with four alpha-helixes, especially enclosing the active site and copper–zinc metal binding loop (residues 49–84) of SOD1 within the protein–protein interface [37]. The SOD1 mutated sites (H46R, G85R, and G93A) were close to the metal binding loop of the enzyme. The protein complex model suggested binding domains between two proteins and provided additional structural-based evidence to explain why these mutations disrupted the PPIs between SOD1 and YWHAE or YWHAZ. The predicted configuration might also explain the PPIs potentially promoted the SOD1 enzymatic activity by protecting the copper–zinc active enzymatic center. Meanwhile, both YWHAE and YWHAZ proteins are widely recognized as adaptor proteins, and no existing methods could directly determine their activity. Our data confer a new, specific function for YWHAE and YWHAZ proteins. However, more actual experiments, including site-directed mutagenesis, co-crystallization, and enzyme kinetics, will be required to explain the mechanism for the enhanced activity of SOD1 by the putative PPIs with YWHAE or YWHAZ.

It is equally exciting for us to demonstrate enhanced stabilities of all three proteins (YWHAE, YWHAZ, and SOD1) by the formation of protein complexes in the PPIs. These enhancements could be attributed to a modified protein-exposed surface when YWHAE or YWHAZ bind with their interactors [46]. Nonetheless, the SOD1 mutants (G85R and D124N) failed to improve the protein stability of YWHAE or YWHAZ, probably due to impaired PPIs between them. Likewise, both SOD1 mutants (G85R and D124N) had lower protein stability than wild-type SOD1 under cycloheximide treatment, consistent with others reported on decreased half-lives (protein stability) on SOD1 mutants (A4V and G85R) [32]. The 14-3-3 family proteins and SOD1 are degraded through the ubiquitin–proteasome system [46,47]. The PPIs between SOD1 and YWHAE or YWHAZ could shield the lysine residues from ubiquitination and ubiquitin-mediated degradation. While the SOD1 mutants disrupted these PPIs, an accelerated breakdown of associated proteins might occur. In the future, the impacts of these PPIs on stability of the involved proteins to resist urea or guanidine treatments should be determined for a comprehensive understanding of protein stability.

The most relevant metabolic implication of our findings is the impact of the PPIs between SOD1 and YWHAE or YWHAZ on lipid metabolism, proliferation, and survival of HEK293T and/or HepG2 cells. SOD1 knockdown (SOD1-KD) induced dysregulated lipid accumulation by altering mRNA levels of SREBP1, LPL, FASN, ACACA, and HMGCR. Because the dysregulation could not be rescued by CuDIPs (SOD1 mimic) treatment, it was likely more related to PPIs of SOD1 rather than its enzymatic function. Remarkably, when an additional knockout of YWHAZ in SOD1-KD cells, the altered mRNA levels of *SREBP1*, *FASN*, and *HMGCR* were restored. It has been reported that SOD1 was involved in DNA repair, bound to RNA in the nucleus, and functioned as a transcription factor [5]. Meanwhile, 14-3-3 family proteins interacted with transcription factors and could mediate their binding to gene promoters [48,49]. Notably, the PPIs between SOD1 and YWHAE or YWHAZ were localized in the nucleus under PCA fluorescent images. Likely, YWHAE and YWHAZ could chaperone SOD1, promote its nuclear localization, and subsequently mediate its regulation of gene expression related to lipid metabolism. The SOD1 mutants (H46R, G85R, and G93A) tested in our study are associated with ALS [31,32,33,34]. It is widely believed that the etiology of these mutations is due to gaining toxic properties by misfolding and aggregation of the mutant SOD1 proteins rather than loss of SOD1 enzymatic activity [34,50]. In fact, overexpressing SOD1-G93A in HepG2 cells in the present study led to decreased stearic acid and oleic acid concentrations and SREBP1, HMGCS2, and LPL mRNA levels, suggesting elevated lipolysis. The loss of *Sod1* was associated with abnormal lipid metabolism [16,17], and *SOD1* human polymorphisms were also linked to incidences of diabetes and obesity [51,52,53,54]. Besides, *Sod1-G93A* mutant mice exhibited hypolipidemia with lowered LDL/HDL ratio and promoted lipolysis with pathological acidosis [55,56], while the *Sod1-G93A* ALS mutant mice exhibited reduced insulin-stimulated glucose uptake in skeletal muscle [57]. Similarly, knockout of YWHAZ also led to increased lipolysis with decreased visceral adipose accumulation [25]. Our discoveries of PPIs’ (SOD1 and YWHAE or YWHAZ) impact on lipid metabolism could provide new clues to those unsettled issues or unexplained phenotypes [25,26,27,55,56,57]. Because 14-3-3 proteins have recently been linked to ALS [58,59,60], our finding on altered cellular lipid metabolism by disrupted PPIs between SOD1 and YWHAE or YWHAZ may reveal a latent factor for the onset and development of ALS.

Similarly, overexpression of YWHAZ is a prognostic biomarker for breast tumors and ovarian cancer, and a reduction in YWHAE levels has been seen in gastric carcinogenesis [61,62,63]. In addition, 14-3-3 family proteins were reported to co-localize with SOD1-A4V aggregates [64], but the impact of the co-localization was unillustrated. Contributions and mechanisms of their PPIs with SOD1 in cancer diseases also remain to be uncovered. Likewise, disrupted or modified PPIs in the SOD1-KD or SOD1-A4V cells might lead to retarded cell proliferation and exacerbated cell death or impaired cell survival. Furthermore, maintaining an appropriate ratio of intracellular SOD1 vs. YWHAE or YWHAZ seemed to be vital for controlling cell growth and survival. Lowering SOD1 activity and protein level (as SOD1-KD and SOD1-A4V) decreased cell growth and survival. Elevating YWHAZ: SOD1 ratio (as SOD1-KD or SOD1-A4V cells transfecting YWHAZ protein) improved cell survival and cell growth, while lowering YWHAZ: SOD1 ratio (as SOD1-KD&YWHAZ-KO overexpressing the SOD1 protein) only enhanced cell proliferation but exacerbated cell death. Furthermore, overproducing and decreasing SOD1 in HEK293T cells fostered and impeded cell proliferation by upregulating or downregulating the *CCND1* gene, respectively. A recent study also discovered elevated SOD1 expression levels in lung cancer cells, facilitating cell proliferation, invasion, and migration [65]. Moreover, YWHAZ is closely associated with cell growth, apoptosis and survival in cancer cells [21,22,66]. A decreased YWHAZ expression inhibited cell proliferation and migration in gastric cancer cells, which frequently detected YWHAZ protein was upregulated [23]. In contrast, a decreased YWHAE expression aggravated tumor growth by potentially inducing cell proliferation, invasion, and migration [67]. The cell growth and survival differences between the SOD1-KD cells overexpressing YWHAE vs. YWHAZ might help explain differential expressions of these two 14-3-3 proteins in tumors, suggesting that PPIs with SOD1 could be a potential modulator of tumorigenesis. Meanwhile, the opposite correlations between YWHAE and YWHAZ with SOD1 from the breast cancer proteome data imply that these PPIs might drive different metabolic impacts in breast cancer development.

The novel perspective of our newly elucidated, unorthodox roles of SOD1 may offer alternative mechanisms to explain paradoxical phenotypes of metabolic disorders. The PPIs could help explain partially a puzzling observation that knockouts of *Sod1* and *Gpx1*, two genes encoding redox enzymes with similar functions in scavenging free radicals, led to different severities in many phenotypes, including hepatic steatosis [17]. Likewise, the PPIs may also help explain that CuDIPs, a SOD mimic, rescued the dysregulated glucose-stimulated insulin secretion in only *Sod1^−/−^* but not *Gpx1^−/−^* islets [42]. Nevertheless, there were several technical limitations in our study. Although we detected the protein complex by GST pull-down assay, immunoprecipitation, and Western blot, the amount of protein complex was low because of the non-specific interacting proteins within the purified YWHAE and YWHAZ proteins. We were not able to perform size-exclusion chromatography and co-crystallization. Further research will be conducted to optimize the condition for forming the protein complex, measuring binding constants between SOD1 and YWHAE or YWHAZ for analyzing the co-crystalline structure to map protein binding domains. Although part of our data was derived from analyses of tissues and cells with stably knocked out or knocked down of genes, many experiments were run with cells overproducing the WT and mutated proteins. Future research is needed to validate the PPIs at physiological conditions of the involved proteins. Although we performed multiple in vitro and in vivo tests and obtained consistent elevations of SOD1 activity due to its interactions with YWHAE or YWHAZ, we hope that our initial findings will lead to more systematic and basic research to characterize the molecular mechanisms and metabolic relevance of the demonstrated PPIs across various cell types, tissues, and species. To distinguish impacts of designated mutations on the enzymatic activity from those on the protein-interaction-based functions, we will prepare mutants depriving the SOD1 activity but maintaining the native structure to evaluate the phenotypes induced by PPIs only. As intracellular PPIs are not necessarily limited to only pairs of two proteins, multi-protein complexes and sophisticated functions should be considered in the future.

Overall, we have elucidated two-way PPIs between SOD1 and YWHAE or YWHAZ and characterized their structural dependences and responses to redox modulation. After revealing the mutual impacts of the PPIs on the enzymatic function (activity) of SOD1 and the protein stability of YWHAE and YWHAZ, we have also demonstrated the effects of the PPIs on lipid metabolism, proliferation, and survival of cultured cells. Overall, our study unveils a new non-canonical role of SOD1 and provides novel perspectives for diagnosing and treating paradoxical diseases related to the protein.

## 4. Materials and Methods

### 4.1. Cells Culture, Animals, Plasmids, and Protein Expression and Purification

HEK293T cells (gifted by Dr. Haiyuan Yu, Cornell University, Ithaca, NY, USA) were maintained in DMEM medium (Cat No. 11995065, Thermo Fisher Scientific, Waltham, MA, USA), and HepG2 cells (gifted by Dr. Ruihai Liu, Cornell University, Ithaca, NY, USA) were maintained in William’s E medium (Cat No. 12551032, Thermo Fisher Scientific, Waltham, MA, USA), supplemented with 10% fetal bovine serum (FBS, Hyclone, Logan, UT, USA) and 1% Antibiotic-Antimycotic (Sigma-Aldrich, Burlington, MA, USA) and incubated, at 37 °C, under air with 5% CO_2_. The experimental cells were cultured with 5–20 passages. The *Sod1^−/−^* and WT mice were generated from the 129SVJ × C57BL/6 strain [68]. Deletion of the *sod1* gene was verified by genotyping using PCR. All mice used in this study were 8-week-old males (*n* = 3–5 replicates). Tissue samples were frozen in liquid nitrogen and stored, at −80 °C, before proceeding to analysis. Our animal experiments were approved by the Institutional Animal Care and Use Committee at Cornell University and conducted following National Institutes of Health guidelines for animal care.

The DNA fragments of human *YWHAZ*, *YWHAE*, and *SOD1*, and murine *Ywhaz, Ywhae*, and *Sod1* (gifted by Dr. Haiyuan Yu, Cornell University, Ithaca, NY, USA) were subcloned into pcDNA3.1-F1 or pcDNA3.1-F2 vectors in N or C terminal (F1N or F1C and F2N or F2C). The cloning and construction of plasmids were performed as described previously [69] or following the Gateway cloning method provided by Thermo Fisher Scientific (Waltham, MA, USA). For constitutive protein expression, *SOD1*, *YWHAE*, and *YWHAZ* were also PCR-amplified from the cDNA library and subcloned into the pPICZαA vector. Sequenced recombinant plasmids were transformed into *Pichia Pastoris X33* yeast, followed by colony PCR confirmation. Positive transformants were grown, and the produced proteins were His-tag purified, following the expression system’s instructions. The *YWHAZ* and *YWHAE* genes were PCR-amplified and subcloned into the pGEX-6P-1 vector (gifted by Dr. Yuxin Mao, Cornell University). The sequenced recombinant plasmids were transformed into *E. coli* strain *BL21 (DE3)* cells, and the transformation was confirmed using colony PCR. The correct transformants were grown, and the produced proteins were purified by GST affinity beads, following the manufacturer’s instructions. Other plasmids used in this study are listed in Appendix A.

### 4.2. Protein Complementation Assay (PCA)

To perform PCA, we seeded HEK293T cells (100 μL, 2 × 10^4^ cells/well) in DMEM media without phenol red (Cat. No 21063029, Thermo Fisher Scientific, Waltham, MA, USA) onto a black 96-well cell culture plate 1 d before transfection (Cat. No 3603, Costar, Washington, DC). On the transfection day, cells were grown to 60–70% confluency and then cotransfected with 100 ng of bait vector (candidate proteins tagged by F2) plus 100 ng of prey vector (target protein SOD1 tagged by F1) or negative control group F1, F2 vectors (Details in Appendix A), using lipofectamine 3000 plus p3000 reagents (Invitrogen™, Carlsbad, CA, USA) mixed with Opti-MEM medium (Cat. No11058021, Thermo Fisher Scientific, Waltham, MA, USA). At 72 h after transfection, fluorescence in the plate was measured using Tecan M1000 spectrophotometer (Zürich, Switzerland) at excitation = 512 nm and emission = 529 nm, or imaged by LSM880 Confocal multiphoton inverted i880 (Core Facilities, Cornell University, Ithaca, NY, USA) with setting laser wavelength 514 nm and detection wavelength 517–642 nm, gain = 650. The measured fluorescent intensity (*n* = 3–5 replicates) was normalized and analyzed against negative vector controls in the same protein terminal orientation. The fold change indicated the relative fluorescence intensity to the background. If the value was close to 1 or lower, there was no PPI. When the value was above 1, the higher the value, the stronger and more abundant the PPIs.

In the oxidative stress assay, diquat and tert-butyl hydroperoxide (TBHP) were used as oxidant generators, and N-acetyl cysteine (NAC), copper diisopropylsalicylate (CuDIPs), and ebselen were used as antioxidants to treat cells at post-transfection. See Appendix A for detailed methods.

### 4.3. Protein Complex Formation and Pull-Down Assay

The GST-fusion protein pull-down was performed as previously described [70,71]. Purified GST-YWHAE or GST-YWHAZ proteins were first run through a 50kDa cutoff centrifuge concentrator (Amicon, Charlotte, NC, USA) to remove non-specific binding proteins. The leftover (>50kDa) GST-YWHAE or GST-YWHAZ protein was immobilized on glutathione sepharose 4B beads (GE Healthcare, Piscataway, NJ, USA), at 4 °C, for 1 h, then rinsed with PBS to remove the non-binding proteins. Thereafter, beads were incubated with purified SOD1-His protein (200 μg) at protein amount ratios of GST-YWHAE or GST-YWHAZ to SOD1-His: 2:1, 1:1, and 1:2, at 4 °C, for 1 h in a pull-down buffer containing 20 mM Tris, 300 mM NaCl, 2 mM β-mercaptoethanol, 1 mM phenylmethylsulfonyl fluoride (PMSF), pH 8.0. After the beads were rinsed with the pull-down buffer 3 times and with PBS 3 times, the eluates were analyzed by SDS-PAGE (12% gel), followed by Coomassie blue staining or Western blot. The SOD1-His, GST-YWHAE or GST-YWHAZ protein was replaced with the same volume of the aforementioned pull-down buffer solution as negative controls.

Since the large molecular weight of GST might block the binding sites, we expressed and purified YWHAZ-His protein via *Pichia Pastoris* system. For immunoprecipitation assay, purified SOD1-His protein (100 μg) was mixed with the anti-SOD1 antibody (2 μg, Appendix A) and incubated, at 4 °C, for 12 h. Subsequently, 50 μL of pre-washed agarose beads G slurry (Sigma, Burlington, MA, USA) was added to the protein–antibody complex solution and made up to 500 μL by binding buffer (20 mM Tris, 150 mM NaCl, 1x protease inhibitor (100x Halt™ Protease Inhibitor Cocktail EDTA-free, Thermo Fisher Scientific, Waltham, MA, USA)) and incubated for 2 h, at 4 °C. The beads were collected by centrifuge and washed 5 times with 500 μL of washing buffer (TBST (20 mM Tris, 150 mM NaCl, 0.1% Tween), 1x protease inhibitor), then centrifuged at 500× *g* for 1 min to remove the supernatant. In the negative controls, the purified SOD1-His protein was replaced by the same volume of binding buffer, or the same amount of rabbit IgG isotype was used to replace the anti-SOD1 antibody. Subsequently, the same amount of purified YWHAZ-His protein (100 μg) was added to the beads to prepare a mixture of 500 μL with fresh-made pull-down buffer (20 mM Tris, pH 8.0, 300 mM NaCl, 2 mM β-mercaptoethanol, 1 mM PMSF) as well as the negative control groups. After incubation at 4 °C for 1 h and 5 times washings with the washing buffer, the beads were eluted with 1x SDS loading buffer, incubated at 70 °C for 10 min, and run for Western blot analysis against the anti-His antibody.

### 4.4. Mutagenesis of the SOD1 Gene and Analyses of the Disrupted PPIs

The mutant human *SOD1* (ALS-related mutants [31,32,33,34]: H46R, G85R, G93A, and zinc active center mutant D124N [35]) gene was generated by Q5^®^ Site-Directed Mutagenesis Kit (New England Biolabs, Ipswich, MA, USA) using the primers listed in Appendix A using the template plasmids WT-SOD1-F1N and WT-SOD1-F1C. These mutants were chosen because: (1) they have different impacts on SOD1 activity, allowing us to test their protein–protein interactions (PPI) with YWHAZ/E at different levels of SOD1 enzymatic function or individually; (2) The AlphaFold structure model suggested these mutations were located in the domain in SOD1 that interacted with YWHAZ or YWHAE; or (3) they (H46R, G85R, and G93A) are involved in ALS, and their PPIs with YWHAZ/E might help explain why SOD1 activity was not the primary etiological factor of ALS.

The sequenced mutagenesis-produced plasmids were used to perform the standard PCA mentioned above. An amount of 100 ng mutant SOD1-F1N or mutant SOD1-F1C was used for co-transfection with F2N vectors (YWHAE-F2N or YWAHZ-F2N) into HEK293T cells. At 72 h after transfection, we measured the whole-plate PCA fluorescence. Co-transfection of WT-SOD1-F1N and WT-SOD1-F1C plasmids with YWHAE-F2N or YWAHZ-F2N in every plate was used for normalization purposes. The measured disruptive mutant interaction is normalized to the percentage change compared with the wild-type interaction, following the Formula (1):(1)$$Relative\;percentage\left(\%\right)\;=\;\frac{Fold\;{\mathrm{change}}_{\mathrm{mutant}}\;-\;1}{Fold\;{\mathrm{change}}_{\mathrm{wild}\;\mathrm{type}}\;-\;1}\;\times \;100\left(\%\right)$$

### 4.5. Measurements of SOD1 and SOD2 Activities

Supernatants were collected from cell pellets lysed in SOD activity buffer (20 mM HEPES, pH 7.2, containing 1 mM EGTA, 210 mM mannitol, and 70 mM sucrose) using an ice sonication cooling cycle. The collected supernatants were diluted (with a buffer of 50 mM Tris-HCL, pH 8.0) or concentrated using a 10 kDa cutoff centrifuge concentrator (Amicon, Charlotte, NC, USA) to adjust enzymatic activity into a linear-curve range. Total SOD activity was measured using a superoxide dismutase assay kit (706002, Cayman, Ann Arbor, MI, USA). The SOD2 activity was measured by adding 10 μL of potassium cyanide solution (freshly prepared) into the sample to a final concentration of 3 mM in the assay to explicitly inhibit the SOD1 activity [72]. The actual SOD1 activity was calculated by subtracting the SOD2 activity from the total SOD activity. In the assay of effect of protein complex formation on SOD1 activity, 20 μg of His-tag-purified SOD1 and YWHAE or YWHAZ protein each were added into 100 μL of the mixture (with the SOD activity assay buffer). The mixture was incubated, at 4 °C, for 0, 1, 2, 4, and 15 h with SOD1 to YWHAE or YWHAZ at 1:1 ratio or for 15 h with SOD1 to YWHAE or YWHAZ at 1:3 ratio. At the end of the incubations, the mixture was used to assay for SOD1 activity.

### 4.6. CRISPR/Cas9 Editing Genome SOD1 and YWHAZ in HEK293T Cells

HEK293T cells (10^6^ seeded on T-25 culture flask) were transiently transfected using lipo3000 with the pX459 vector (gifted by Dr. John Schimenti, Cornell University, Ithaca, NY, USA) integrated with the guide RNA (gRNA) targeting 5′- CTAGCGAGTTATGGCGACGA-3′ sequence in exon 1 of the *SOD1* gene (as pX459-SOD1), or pDG459 (gifted by Dr. Tudorita Tumbar, Cornell University, Ithaca, NY, USA) integrated the dual gRNAs targeting 5′-CATGACTGGATGTTCTGCAG-3′ and 5′-AGATATCTGCAATGATGTAC-3′ sequences in exon 4 of the *YWHAZ* gene (as pDG459-YWAHZ) following standard cloning method described in the protocol [73,74] to generate stable SOD1 knockdown (SOD1-KD) or YWHAZ knockout (YWHAZ-KO) cells. The SOD1-A4V (alanine to valine at fourth position) genome mutation was created by homologous recombination repairing of CRISPR-Cas knock-in through adding single-stranded oligodeoxynucleotides (ssODNs) as a donor template during the pX459-SOD1 transfection period. We only screened out SOD1-KD cells because SOD1 knockout seemed lethal to our HEK293T cells. We constructed the SOD1-A4V mutant in the cell line because the A4V mutant is also related to ALS, which depleted SOD1 activity by 97% (Appendix A) and is the most common mutant with rapid disease progression in US ALS patients [75]. In this way, we could generate both SOD1-KD and SOD1-A4V as protein-loss and ALS-related cell lines to test intracellular conditions under impaired SOD1 protein and activity.

Primer sets used for the annealing formation of 20 bp gRNA oligonucleotides were listed in Appendix A. At 48 h after transfection, cells were selected using puromycin (0.8 μg/mL) for another 72 h. The cells were harvested, washed 3 times with PBS, and resuspended in FACS buffer (1L PBS, 1g BSA, 2mM EDTA) to a cell density of 10^6^. The cells were run through SONY MA900 flow cytometry (Core Facility, Cornell University, Ithaca, NY, USA) and sorted out as single cells into the 96-well plate. After the single colony grew, part was used to extract DNA by an alkaline lysis method [76], followed by colony PCR (primers in Appendix A) and sequencing of exon 1 of the *SOD1* gene or exon 4 of the *YWHAZ* gene. The rest of the colony was maintained under standard culture and passage. The heterozygotes could continue to perform the TA cloning (TOPO™ TA Cloning™ kit, Thermo Fisher Scientific, Waltham, MA, USA) on colony PCR products to confirm the sequence information for each chromosome. Successfully edited colonies were maintained for subsequent experiments. The SOD1 protein knockdown cells were further transfected with the pDG459-YWHAZ vector and followed the same process to select SOD1 protein knockdown and YWHAZ protein knockout cells.

### 4.7. Co-Immunoprecipitation

HepG2 cells (seeded on T-25 culture flasks) were transfected with human WT-SOD1-F1N, SOD1-H46R-F1N, SOD1-G93A-F1N, or F1N as a control vector. After transfection for 72 h, the cells were harvested and lysed in 400 μL of lysis buffer (25 mM Tris-HCl, pH 7.4, 150 mM NaCl, 1 mM EDTA, 1% NP-40, 5% glycerol). Cell lysates (1 mg protein) were incubated with 2 μg of SOD1 antibody (Santa Cruz Biotechnology, Dallas, TX, USA) for 16 h, at 4 °C. The Pierce Protein A Magnetic beads (100 μL per sample, Thermo Fisher Scientific, Waltham, MA, USA) were pre-washed by a washing buffer (25 mM Tris-HCl, pH 7.4, 0.5 M NaCl, 0.05% Tween) following the user guide. Afterward, the antigen sample/antibody mixture was incubated with pre-washed magnetic beads, at room temperature, for 1 h in an end-to-end rotator. The beads were washed for 3 times with the washing buffer and 1 time with ultrapure water, and eluted into 80 μL of SDS-PAGE sample loading buffer boiled at 90 °C for 10 min. The supernatants were used for SDS-PAGE analysis and immunoblotting against SOD1, YWHAE, and YWHAZ primary antibodies and a specific IP detection secondary antibody to minimize the background (VeriBlot, ab131366, Abcam, Waltham, MA, USA).

### 4.8. Protein Degradation Assay

The cycloheximide (Sigma, Burlington, MA, USA) chase assay was applied for the protein degradation assay, where cycloheximide could entirely block protein synthesis [77]. It is also called a protein stability assay described in the previous paper [46], indicating the protein’s resistance to itself degradation. HEK293T cells (seeded on 12-well plates overnight) were cotransfected with SOD1-F1N or mutant SOD1-F1N (G85R and D124N) and YWHAE-F2N or YWHAZ-F2N, or F1N or F2N control vectors as negative controls, respectively. After 24 h of transfection, cells were treated with freshly prepared cycloheximide (20 μg/mL) or PBS as the control. The cells were harvested at 0 or 24 h after the treatment to determine relative protein amounts of YWHAE, YWHAZ, and SOD1 by Western blot. The protein stability was reflected by quantifying protein band intensity of 24 h normalized to its 0 h, respectively.

### 4.9. Staining of Lipid Droplets and Detection of Lipid Profiles

For lipid droplets staining, HepG2 cells were seeded onto a black 96-well cell culture plate 1 d before transfection. The cells were transfected with WT-SOD1-F1N, mutant SOD1-F1N (H46R and G93A), and F1N control vector, or cotransfected with YWHAE-F2N and SOD1-F1N or F1N control vector. The lipid droplets in the cells were stained using Nile Red Staining Kit (Abcam, Waltham, MA, USA). After 48 h of transfection, culture media was aspirated before the addition of 100 μL of Nile Red Staining solution per well. Cells were incubated, at 37 °C, under air with 5% CO_2_ for 30 min. The whole plate was measured with Nile red fluorescence using Tecan M1000 spectrophotometer (Zürich, Switzerland) at excitation = 550 nm and emission = 640 nm, or the cells were imaged using LSM880 Confocal multiphoton inverted i880 (Core Facilities, Cornell University, Ithaca, NY, USA) with setting laser wavelength excitation = 514 nm, emission = 646 nm, detection wavelength 539–753 nm, gain = 532. The WT and CRISPR-genome-edited SOD1-KD (SOD1 knockdown) and SOD1-KD&YWHAZ-KO (SOD1 knockdown and YWHAZ knockout) cells (10^4^ per well) were plated in black 96-well plates for 24 h and were treated with CuDIPs (1 or 10 μM of final concentrations) or an identical amount of DMSO vehicle. After the treatment for 48 h, cell culture media was aspirated before the addition of 100 μL of Nile Red Staining solution per well. Cells were incubated, at 37 °C, under air with 5% CO_2_ for 30 min, followed by standard Nile red fluorescence measurement or fluorescence microscope imaging mentioned above.

Total cholesterol (TC), total triglycerides (TG), and non-esterified fatty acid (NEFA) in HepG2 cells (10^6^ cells/flask) were determined at 72 h after transfection. Cells were sonicated in 500 μL of PBS, and the supernatant protein concentration was determined using the Pierce Bicinchoninic Acid (BCA) protein assay kit (Thermo Fisher Scientific, Waltham, MA, USA). The remaining supernatant (450 μL) was mixed with 1 mL of chloroform: isopropanol: NP-40 = 7:11:0.1 solution and centrifuged. The lower layer was transferred into a glass tube, dried under nitrogen gas, and dissolved into 200 mL of ethanol containing 1% Triton. Concentrations of TC, TG, and NEFA in the ethanol solution were measured using kits (Wako Chemicals, Richmond, VA, USA). Fatty acid profiling in the cell homogenates, along with lipid extraction, was performed as described previously [78].

### 4.10. Cell Growth and Survival

HEK293T (WT), SOD1-A4V, SOD1-KD (SOD1 knockdown), and SOD1-KD&YWHAZ-KO (SOD1 knockdown and YWHAZ knockout) cells (10^4^ cells/well) were seeded in 96-well plates overnight and transfected with 100 ng of SOD1-F1N, YWHAZ-F2N, YWHAE-F2N or their respective vector controls (F1N, F2N) using lipofectamine 3000 plus p3000 reagents.

After 48 h of transfection, cells were further cultured in 100 μL of fresh DMEM supplemented with 10% FBS for 0, 24, or 48 h for proliferation assay. At each time-point, 10 μL of MTT (12 mM) solution was added into each well to measure cell growth following a standard CyQUANT MTT cell assay protocol by Thermo Fisher Scientific (Waltham, MA, USA). Following incubation with MTT for 4 h, at 37 °C, 100 μL SDS-HCl (1 g SDS in 10 mL 0.01 M HCl) was added to each well. Incubated the microplate, at 37 °C, for another 4 h to completely dissolve the blue crystal formed. Mixed each sample thoroughly by pipetting up and down. The final absorbance was then measured at 570 nm using the SpectraMax M2e spectrophotometer (Silicon Valley, CA, USA).

Cell survival was also determined using the MTT with the assay previously described [79]. After 48 h of transfection, cells were cultured and stressed in 100 μL of DMEM without FBS for 24, 48, or 72 h before adding 10 μL of MTT (12 mM) to each well. Cell survival was set as 100% at the initial time point and relative survival was calculated for each time point thereafter.

### 4.11. Quantitative Real-Time PCR (qPCR)

Total RNA was extracted from cells (~10^6^ cells) using an RNeasy Mini kit (QIAGEN, Germantown, MD, USA). The high-capacity cDNA reverse transcription kit (Cat No. 43-688-14, Applied Biosystems, Waltham, MA, USA) was used for reverse transcription. The Real-time qPCR used an iTaq Universal SYBR Green Supermix (Cat No. 1725124, Bio-Rad Laboratories, Hercules, CA, USA) according to the manufacturer’s instructions on QuantStudio 7 Pro (Thermo Fisher Scientific, Waltham, MA, USA), and 2^–delta delta Ct (∆∆Ct)^ equation [80] was used to quantify relative mRNA levels normalized to levels of the housekeeping gene *GAPDH*. Primers used for qPCR are listed in Appendix A.

### 4.12. Western Blotting

Cells (10^6^ cells) or mouse tissues (50mg) were lysed in RIPA buffer with 1x Halt™ Protease Inhibitor Cocktail, EDTA-free (Thermo Fisher Scientific, Waltham, MA, USA). Protein concentration was assayed by the BCA protein assay kit described above. Supernatants (40–60 µg of protein per lane) of the lysates were subjected to Western blot analyses as described previously [81]. Relative densities of protein bands in individual blots were quantified using SuperSignal West Pico PLUS Chemiluminescent Substrate (Thermo Fisher Scientific, Waltham, MA, USA), and the membrane was imaged by ChemiDoc^TM^ MP Imaging System (Bio-Rad). In addition, all immunoblot band intensities in this study were analyzed by ImageJ (National Institutes of Health, Bethesda, MD, USA). Antibodies used in this study are listed in Appendix A.

### 4.13. Correlation Analysis of the Protein Expression Levels

The protein expression data were acquired from a breast cancer quantitative proteome database (Henrik J. Johansson, 2019) [41]. The abundance of the proteins was further processed by the tools suggested by Dr. Nathaniel Vacanti (Cornell University, Ithaca, NY, USA). The final protein data was relative to an internal standard, log_2_ transformed, and then standardized to z-score. The missing data and outliers were removed during data processing. The *x*-axis was set as SOD1, and the *y*-axis was set as YWHAE or YWHAZ. Pearson correlation coefficient (r) with *p*-value was calculated for SOD1-YWHAE and SOD1-YWHAZ pairs.

### 4.14. Statistical Analysis

Data are presented as mean ± SE (*n* = 3–6). Data generated from experiments with more than 2 treatment groups were analyzed with R software (version 4.0.3, R Core Team, Vienna, Austria) using one-way ANOVA followed by Duncan’s test. Data generated from experiments with only 2 treatment groups were analyzed using Student’s *t*-test. The Pearson correlation was analyzed with GraphPad Prism 8.0.1 (GraphPad Software, Inc., San Diego, CA, USA). Statistical significance of differences was set at and indicated as *: *p* < 0.05; **: *p* < 0.01; and ***: *p* < 0.001.

## Figures and Tables

**Figure 1 ijms-24-03230-f001:**
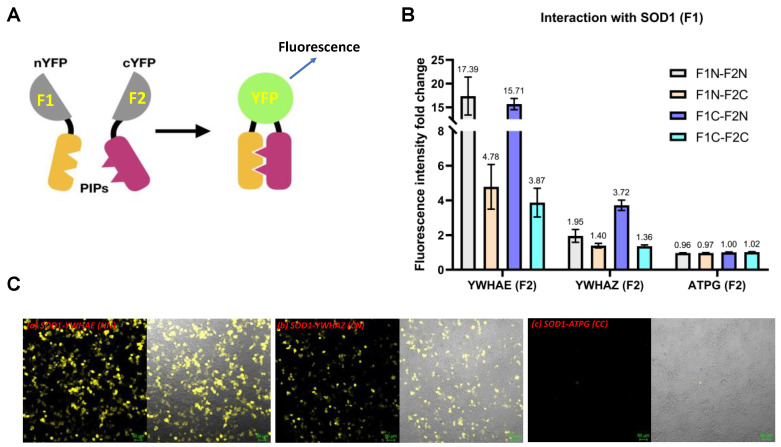
Evidence for protein–protein interactions between human SOD1 and YWHAZ or YWHAE by protein complementation assay (PCA). (**A**) A scheme of the PCA principle. SOD1 was tagged with F1, and YWHAZ, YWHAZ, and ATPG were tagged with F2. (**B**) Relative binding intensity between SOD1 and YWHAZ, YWHAE, or ATPG at different orientations. The fold change of fluorescent intensity was normalized to the control vector pair in the same orientation as a background control. (**C**) Representative images of fluorescent microscopy of transiently transfected HEK293T cells expressing the target protein pairs: (**a**) SOD1-YWHAE in F1N-F2N, (**b**) SOD1-YWHAZ in F1C-F2N, (**c**) SOD1-ATPG in F1C-F2C. Scale bar, 50 μm. The left panel is a fluorescence image, and the right panel is the fluorescence overlapping with its brightfield image. The image was representative of three sets of data.

**Figure 2 ijms-24-03230-f002:**
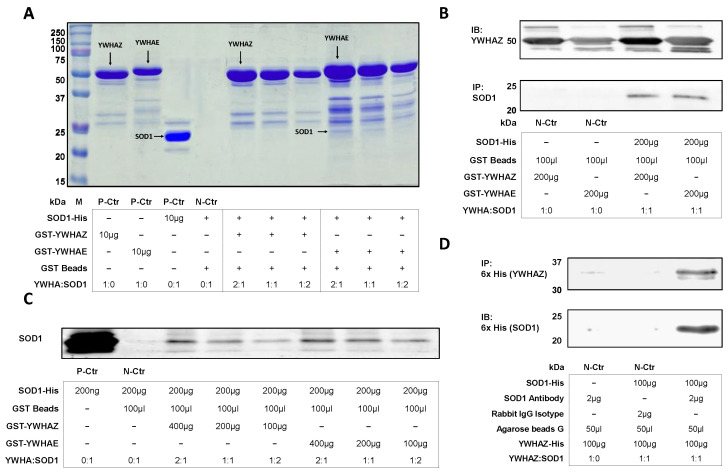
Evidence for protein–protein interaction complex formation of YWHAZ-SOD1 and YWHAE-SOD1. (**A**) A representative SDS-PAGE image with Coomassie blue staining showed the protein complex formation between purified GST-YWHAZ or GST-YWHAE immobilized on glutathione sepharose beads and the purified SOD1-His at different ratios. P-Ctr: positive control. N-Ctr: Negative control. (**B**) A representative image of immunoblots of YWHAZ and SOD1 with samples of the same protein complexes (GST-YWHAZ&SOD1-His or GST-YWHAE&SOD1-His) eluted from the glutathione sepharose beads. (**C**) A representative image of immunoblots of SOD1 with the same protein complexes (GST-YWHAZ&SOD1-His, or GST-YWHAE&SOD1-His) eluted from the glutathione sepharose beads at different ratios. (**D**) A representative image of the immunoprecipitation pull-down showed the binding protein complex formation between purified SOD1-His with anti-SOD1 antibody immobilized on agarose beads G. The eluates were subjected to SDS-PAGE (12% gel) and immunoblotted by the anti-His antibody.

**Figure 3 ijms-24-03230-f003:**
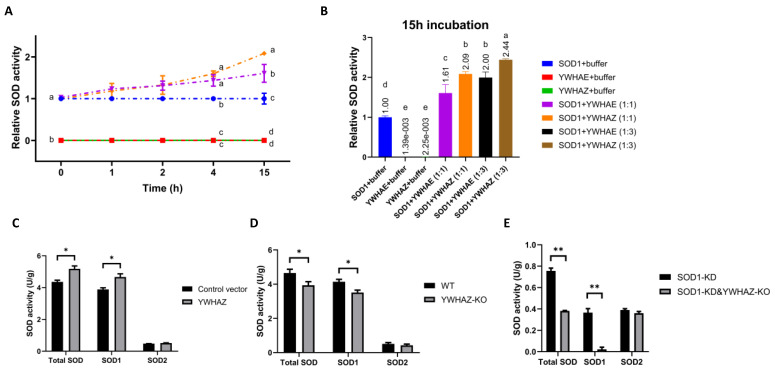
Effects of protein–protein interactions (PPIs) between SOD1 and YWHAE or YWHAZ on SOD1 activity. (**A**) The SOD1 activity in the mixture of SOD1-YWHAE or SOD1-YWHAZ at different incubation time points. The time-point relative SOD activity value was normalized to each time-point “SOD1 + buffer” group’s value. SOD1, YWHAE, and YWHAZ proteins were overexpressed in *Pichia pastoris* X33 and purified by His-tag affinity beads (*n* = 2). (**B**) The SOD1 activity in the mixture of SOD1-YWHAE or SOD1-YWHAZ at different protein ratios for 15 h incubation (*n* = 2). (**C**) The total SOD, SOD1, and SOD2 activities of HEK293T cells transfected with the YWHAZ-F2N vector compared with the F2N control vector (*n* = 3). (**D**) The total SOD, SOD1, and SOD2 activities of YWHAZ knockout HEK293T cells compared with wild-type HEK293T cells (*n* = 3). (**E**) The total SOD, SOD1, and SOD2 activities of YWHAZ knockout and SOD1 knockdown cells compared with the parent SOD1 knockdown cells (*n* = 3). *: *p* < 0.05 and **: *p* < 0.01. Means within a given time (**A**) and in (**B**) without sharing a common letter differ (*p* < 0.05).

**Figure 4 ijms-24-03230-f004:**
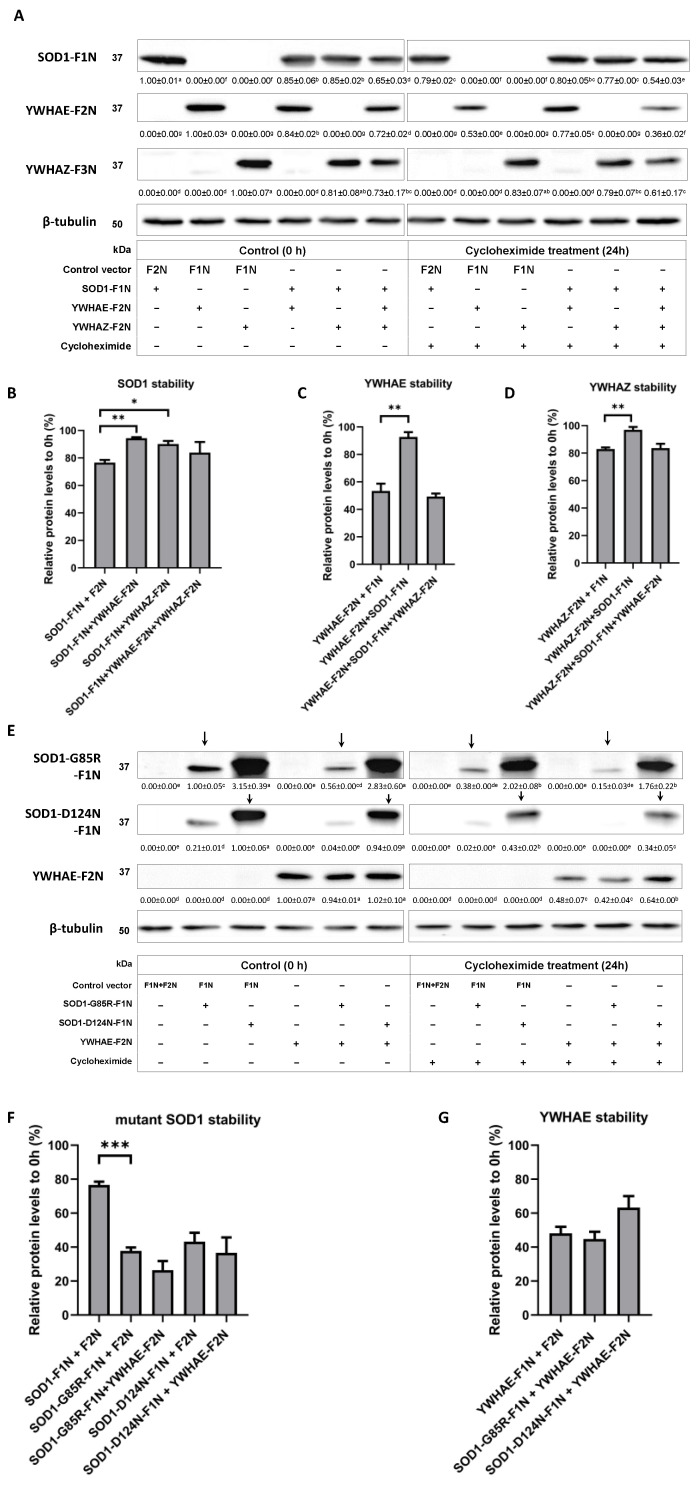
Effects of protein–protein interactions between YWHAE or YWHAZ with native and variant SOD1 proteins on their protein stability in HEK293T cells. (**A**) Immunoblots of protein stability in cells transfected with plasmids expressing WT-SOD1, YWHAE, and YWHAZ. Cells were treated with 20 μg/mL cycloheximide for 24 h after transfection. The cell lysates were probed with anti-SOD1, anti-YWHAE, and anti-YWHAZ antibodies (*n* = 2). The values were normalized to the expression levels of β-tubulin. (**B**) Quantification of protein stability of WT-SOD1 in cells cotransfected with WT-SOD1-F1N and YWHAE-F2N or YWHAZ-F2N or respective vector controls. The results were normalized to 0 h protein expression level (*n* = 2). (**C**) Quantification of protein stability of YWHAE in cells cotransfected with YWHAE-F2N and WT-SOD1-F1N or respective vector controls. The results were normalized to 0 h protein expression level (*n* = 2). (**D**) Quantification of protein stability of YWHAZ in cells cotransfected with YWHAZ-F2N and WT-SOD1-F1N or respective vector controls. The results were normalized to 0 h protein expression level (*n* = 2). (**E**) Immunoblots of protein stability in cells transfected with plasmids expressing SOD1 mutants (G85R and D124N) and YWHAE. Cells were treated with 20 μg/mL cycloheximide for 24 h after transfection. The cell lysates were probed with anti-SOD1 and anti-YWHAE antibodies (*n* = 2). The values were normalized to the expression levels of β-tubulin. (**F**) Quantification of the protein stability of YWHAE in cells cotransfected with mutant SOD1-F1N (G85R and D124N) and YWHAE-F2N or respective vector controls. The results were normalized to their 0 h protein expression level (*n* = 2). (**G**) Quantification of the protein stability of YWHAE in cells cotransfected with YWHAE-F2N and WT-SOD1-F1N or respective vector controls. The results were normalized to their 0 h protein expression level (*n* = 2). *: *p* < 0.05; **: *p* < 0.01; and ***: *p* < 0.001.

**Figure 5 ijms-24-03230-f005:**
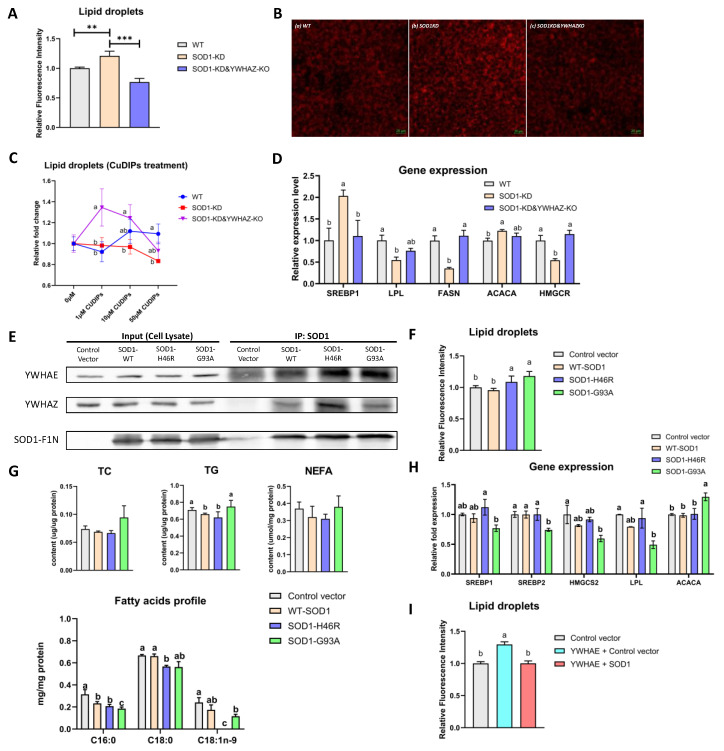
Impacts of the disrupted SOD1 and YWHAZ or YWHAE protein–protein interactions on cellular lipid metabolism in SOD1 and YWHAZ genome-edited HEK293T cells or HepG2 cells overexpressing SOD1 mutants. (**A**) Quantification of lipid droplets in vivo staining by Nile red assay in WT, SOD1-KD, and and SOD1-KD&YWHAZ-KO HEK293T cells. (**B**) Fluorescent images of Nile red-stained genome-edited cells. *Annotation*: (**a**): WT (Wild type), (**b**): SOD1-KD (SOD1 knockdown), (**c**): SOD1-KD&YWHAZ-KO (SOD1 knockdown and YWHAZ knockout), Scale bar, 20 μm. The image was representative of three sets of data. (**C**) Quantification of lipid droplets in vivo staining by Nile red assay under treatments of graded levels of CuDIPs for 48 h in WT, SOD1-KD, and SOD1-KD&YWHAZ-KO HEK293T cells. (**D**) Relative mRNA levels of lipid metabolism genes WT, SOD1-KD, and SOD1-KD&YWHAZ-KO HEK293T cells (*n* = 4). (**E**) Co-immunoprecipitation of YWHAE and YWHAZ pulled down by SOD1 antibody in HepG2 overexpressing WT-SOD1-F1N, mutant SOD1-F1N (H46R and G93A), or F1N control vector. The cell lysates were used as positive controls for the Western blot. (**F**) Quantification of lipid droplets in vivo staining by Nile red assay in HepG2 cells overexpressing control vector (F1N), WT-SOD1-F1N, mutant SOD1-F1N (H46R and G93A), or co-overexpressing WT-SOD1-F1N and YWHAE-F2N or YWHAE-F2N and F1N control vector (*n* = 3). (**G**) TC, TG, NEFA concentrations, and fatty acids profiles in the lysates of HepG2 cells overexpressing WT-SOD1, SOD1 mutants (H46R and G93A), or control vector F1N (*n* = 3). (**H**) Relative mRNA levels of lipid metabolism genes in HepG2 cells overexpressing control vector F1N, WT-SOD1, or SOD1 mutants (H46R and G93A) (*n* = 4). **: *p* < 0.01 and ***: *p* < 0.001. Means for a given treatment (dose, gene, molecule, or fluorescence intensity) without sharing a common letter differ (*p* < 0.05).

**Figure 6 ijms-24-03230-f006:**
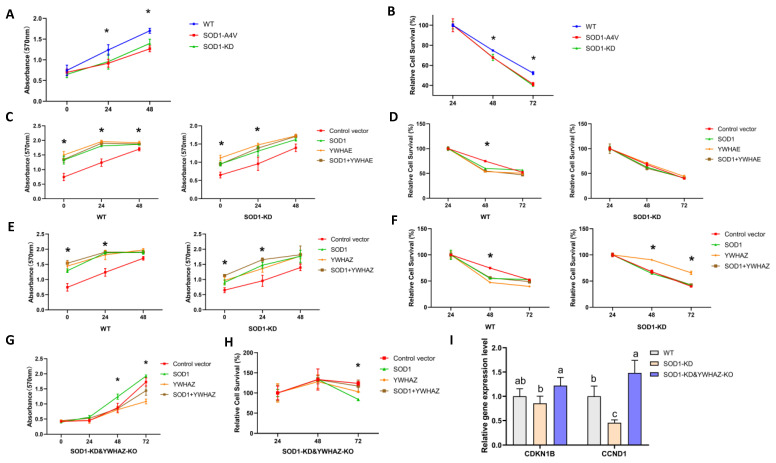
Impacts of altering protein–protein interactions between SOD1 and YWHAE or YWHAZ on cell growth and survival of HEK293T cells. (**A**) Proliferation curves in WT, SOD1A4V, and SOD1-KD HEK293T cells. (**B**) Survival curves of WT, SOD1A4V, and SOD1-KD HEK293T cells after the removal of FBS in culture media. (**C**) Proliferation curves of WT and SOD1-KD cells overexpressing SOD1 and YWHAE alone or in combination supplemented with FBS culture media. (**D**) Survival curves of WT and SOD1-KD cells overexpressing SOD1 and YWHAE alone or in combination after the removal of FBS in culture media. (**E**) Proliferation curves of WT and SOD1-KD cells overexpressing SOD1 and YWHAZ alone or in combination supplemented with FBS culture media. (**F**) Survival curves of WT and SOD1-KD cells overexpressing SOD1 and YWHAZ alone or in combination after the removal of FBS in culture media. (**G**) Proliferation curves of SOD1-KD&YWHAZ-KO cells overexpressing SOD1 and YWHAZ alone or in combination supplemented with FBS culture media. (**H**) Survival curves of SOD1-KD&YWHAZ-KO cells overexpressing SOD1 and YWHAZ alone or in combination after the removal of FBS in culture media. (**I**) Relative mRNA levels of cell proliferation genes in SOD1-KD and SOD1-KD&YWHAZ-KO cells compared with WT cells. *: The asterisks above curves indicated that the SOD1A4V and SOD1-KD were significantly different from the WT in (**A**,**B**). In (**C**–**H**), the asterisks indicated that at least one treatment group was significantly different from the negative control (F1N + F2N), *p* < 0.05, *n* = 4. Means for a given gene (**I**) without sharing a common letter differ (*p* < 0.05).

**Figure 7 ijms-24-03230-f007:**
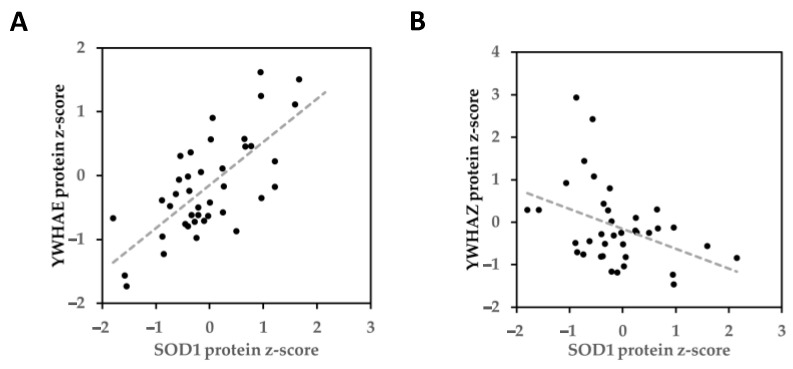
The correlation analysis on SOD1-YWHAE and SOD1-YWHAZ. (**A**) The scatter plot by standardized protein z-score of SOD1 versus YWHAE based on breast cancer proteomic dataset, r = 0.71, *p*-value = 3.05 × 10^−7^. (**B**) The scatter plot by standardized protein z-score of SOD1 versus YWHAZ based on breast cancer proteomic dataset, r = −0.40, *p*-value = 0.015.

**Table 1 ijms-24-03230-t001:** Impacts of SOD1 mutants on its interactions with YWHAE and YWHAZ.

Candidates	SOD1		SOD1-Mutants							
	WT		H46R		G85R		G93A		D124N	
	NN	CN	NN	CN	NN	CN	NN	CN	NN	CN
Fold change (To negative control)										
YWHAE	6.53	14.7	2.99	3.86 *	1.50 *	4.25 *	2.47 *	1.60 **	1.60 *	1.54 **
YWHAZ	2.59	3.52	1.32 *	1.72 **	1.01 *	2.04 **	1.06 *	1.50 **	1.08 *	2.27 **
Relative percentage (%, to wild type)										
YWHAE	100	100	36.0	20.9	8.99	23.8	26.6	4.35	10.8	3.98
YWHAZ	100	100	20.0	28.7	0.33	41.4	3.67	19.7	5.02	50.3

NN: SOD1 or SOD1 mutants tagged with F1 in N-terminal interacted with YWHAE or YWHAZ tagged with F2 in N-terminal. CN: SOD1 or SOD1 mutants tagged with F1 in C-terminal interacted with YWHAE or YWHAZ tagged with F2 in N-terminal. The fold change was normalized to negative control F1N-F2N or F1C-F2N as 1.00. The relative percentage converted the SOD1 mutants fold change to a percentage relative to SOD1-WT interacting with YWHAE or YWHAZ by Formula (1) in the method section. The statistical analysis compared each SOD1 mutant with WT-SOD1 interacting with YWHAE or YWHAZ in the same orientation (NN or CN) by Student’s *t*-test. *: *p* < 0.05; and **: *p* < 0.01. Data are presented as mean (*n* = 3).

## Data Availability

Not applicable.

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
