# Peer review of "Evidence and Metabolic Implications for a New Non-Canonical Role of Cu-Zn Superoxide Dismutase"

_ijms, 2023, doi:10.3390/ijms24043230_

Round 1

Reviewer 1 Report

This is an extensive study that investigates a non canonical function of SOD1 in the regulation of lipid metabolism via interactions with YWHAE, and or YWHAZ in multiple cell lines including HEK293 T cells, HepG cells, cancer cells and mice. However, the effect of sod1 dysregulation was not examined in obesity, diabetes or growth retardation, where lipid metabolism is likely derranged.

Here are my comments:

Major comments

1) The authors examined the effect of sod1 mutations(H46R, G85R, G93A, D124N) on interactions with YWHAZ or YWHAE. They did not explain the significance of these mutations on SOD1 activity. Are these mutations in the domain in sod1 that interacts with YWHAZ and or YWHAE? Later on , they tested the effect of another mutation sod1 A4). What is the relationship of this mutation on sod1-YWHAZ/YWHAE interactions? if i understood the premise of this study, which is to scrutinize sod1 interactions with YWHAZ /YWHAE. If so, justice is not done to define the detail mechanism of these interactions without knowing the interactions domains. Moreover, the explanation using crystalline structure did not reveal Sod1 interaction with YWHAE or YWHAZ.

2) Re-expression of YWHAE and SOD1 alone or together in sod1 KO cells rescued cell growth but nit cell viability. Can non viable cells grow?

3) Treatment with diquat, and TBHP attenuated the interaction at 1 and 2 h but Figure S2 indicate the reverse. Can you reconcile? Further, since sod1 is involved in lipid metabolism, no document of sod1 mutations in obesity or diabetes.

4) The authors suggest that sod1 interactions with YWHAE increased lipolysis by increasing mRNA levels of several genes. Can the author explain how? Studies have shown that sod1 is involved in DNA repair and also binds to RNA in the nucleus. Could it be that YWHAE or YWHAZ chaperones and promotes sod1 nuclear localization and gene expression? The increased sod1 antioxidant activity through its interaction with these adaptors can not explain the mechanism of increased expression of lipoprotein genes.

Minor

1) All the blots have no background. Can you adjust the contrast?

2) The author did not include interpretation of results in the result sections making it difficult to understand the manuscript.

Author Response

Responses are submitted in a separate file

Reviewer 2 Report

Sun et al. provided more results mainly through PCA and pull down assays to further confirm the interaction between SOD1 and YWHAZ or YWHAE indicated by AP-MS in the previous work, and explored the cellular effect of these PPIs. The following concerns would be considered in the modified version of this submission.

1.     These authors should fully describe the non-canonical roles of SOD1, e.g, the DNA binding of SOD1 and its regulation roles in gene expression.

2.     The statistical differences in PCA results in Figure 1 were analyzed by comparison with ATPG in the same orientation. This analysis is meaningless because ATPG does not interact with SOD1.

3.     The small bands in Figure 2A indicate the presence of impurity proteins in the proteins of interest samples. It is impossible that SDS-PAGE provides evidence for the formation of protein complexes, but MS determinations of the cross-linked protein mixtures might indicate the formation of SOD1-YWHAZ or –YWHAE complexes. In order to directly confirm formation of the SOD1 protein complexes, the authors should also quantitatively give the binding constants of SOD1 to YWHAZ or YWHAE in solutions by using changes in the YFP fluorescence intensity or by other determination methods.

4.     Figure s3 shows that overexpressing these mutants of SOD1 did not alter intracellular SOD1 activity. Do these mutants of SOD1 not provide any contribution to total intracellular SOD1 activity?

5.     The authors explain why the interaction with YWHAZ or YWHAE can elevate SOD1 activity by using the structural models in Figure s8. Obviously, this explanation is unreasonable. Here, evidence from experiments is needed to explain the elevated SOD 1 activity.

6.     The relative protein levels of SOD1 are indicated in Figure 4. Protein levels are not equal to protein stability. The resistance to urea or guanidine treatment of proteins can indicate changes in protein stability. Obviously, here, evidence from experiments is needed to explain the elevated SOD 1 protein level.

7.     The interaction with YWHAZ or YWHAE can elevate both activity and protein level of SOD1. These unbelievable results need to be supported by more experimental determinations.

8.     It is difficult to recognize the trends and differences of data in Figure 5.

9.     I do not understand significance of both tests in sections 2.7 and 2.8.

Author Response

please refer to submitted response file (2)

Round 2

Reviewer 1 Report

I have no further question.

Author Response

We thank Reviewer 1 for accepting our revised manuscript.

Reviewer 2 Report

The authors have addressed most of my concerns. However there is still several concerns that would be considered in the modified version of this submission.

1.     The authors should cite more recently published references .

2.    The authors should  quantitatively give the binding constants of SOD1 to YWHAZ or YWHAE in solutions via fluorescence  or MST assays.

Author Response

We thank Reviewer 2 for constructive questions and have responded in our attached document.
